# Predicting Potential SARS-COV-2 Drugs—In Depth Drug Database Screening Using Deep Neural Network Framework SSnet, Classical Virtual Screening and Docking

**DOI:** 10.3390/ijms22041573

**Published:** 2021-02-04

**Authors:** Nischal Karki, Niraj Verma, Francesco Trozzi, Peng Tao, Elfi Kraka, Brian Zoltowski

**Affiliations:** Department of Chemistry, Southern Methodist University, Dallas, TX 75205, USA; nkarki@smu.edu (N.K.); nirajv@smu.edu (N.V.); ftrozzi@smu.edu (F.T.); ptao@smu.edu (P.T.); ekraka@smu.edu (E.K.)

**Keywords:** drugs for SARS-COV-2, coronavirus, deep neural network, SSnet, docking

## Abstract

Severe Acute Respiratory Syndrome Corona Virus 2 has altered life on a global scale. A concerted effort from research labs around the world resulted in the identification of potential pharmaceutical treatments for CoVID-19 using existing drugs, as well as the discovery of multiple vaccines. During an urgent crisis, rapidly identifying potential new treatments requires global and cross-discipline cooperation, together with an enhanced open-access research model to distribute new ideas and leads. Herein, we introduce an application of a deep neural network based drug screening method, validating it using a docking algorithm on approved drugs for drug repurposing efforts, and extending the screen to a large library of 750,000 compounds for de novo drug discovery effort. The results of large library screens are incorporated into an open-access web interface to allow researchers from diverse fields to target molecules of interest. Our combined approach allows for both the identification of existing drugs that may be able to be repurposed and de novo design of ACE2-regulatory compounds. Through these efforts we demonstrate the utility of a new machine learning algorithm for drug discovery, SSnet, that can function as a tool to triage large molecular libraries to identify classes of molecules with possible efficacy.

## 1. Introduction

Pathogens have wreaked havoc on human society for as long as human society has existed. Various forms of pathogenic microbes have marked pivotal points in human history among which the notable examples are plague, smallpox, tuberculosis, and cholera [1]. While most of these pathogens have been either eradicated or have a cure developed, as globalization increases, new emergent diseases remain an increasing global threat. Ebola virus, hantavirus, zika virus, human immunodeficiency virus, and coronaviruses are some of the viral families that have been identified and have continuously posed a threat in the past decades [2,3]. Currently, we are amidst a pandemic caused by a member of the coronavirus family, Severe Acute Respiratory Syndrome Coronavirus-2 (SARS-CoV-2) which has claimed 1.7 million lives and significantly impacted the global economy. The pharmaceutical research in response has been relentless and fruitful, however as seen during the ongoing pandemic, despite recent technology breakthroughs, development and widescale production of vaccines take well over a year. As a result, it is imperative that we develop rapid methods to identify putative therapeutics to combat future rounds of new emergent diseases.

Disease prevention and treatment can proceed via three avenues with varying utility and timelines. Long-term, a viable vaccine is the best option for an intervention strategy to mitigate the spread and effect of a virus. Short-term, drug re-purposing from approved drug data-sets is most effective, since they can be deployed as soon as they show efficacy against the disease. de novo development of a targeted treatment would require years of testing and regulatory approval, thus, although such a development route may be higher in terms of efficacy, the timeline precludes rapid usage of such an approach. Vaccine development is a difficult process and thus can potentially take much longer to be developed where highly specific de novo drugs could bridge the timeline. For these reasons, the research and pharmaceutical community must focus on a three-tiered platform for disease prevention, treatment, and eventually eradication of emergent diseases.

Against SARS-CoV-2, scientists throughout the world have pushed research on identifying epidemiology, drug re-purposing, de novo drug design, and development of vaccines. With over two thousand clinical studies registered in www.clinicaltrials.gov, many researchers have identified various means of prevention and treatments. Towards eradication, vaccines have been developed with 56 currently in clinical trial and 146 in pre-clinical development worldwide [4]. Multiple vaccines have been authorized for use by the Food and Drugs Administration (FDA) in the USA which has projected over 100 million doses delivered worldwide by March of 2021 [5]. Furthermore, drug re-purposing has allowed the use of many drugs, particularly antivirals for relief against severe disease progression. The rapid developmental success of vaccine has lessened the urgency for de novo drugs that target SARS-CoV-2 with high affinity. Through this diversified focus on dealing with SARS-CoV-2, researchers have managed to find effective means of reducing the severity of cases and most importantly development of several versions of vaccine against SARS-CoV-2.

SARS-CoV-2, like all members of coronavirus family, has a crown-like spike protein (S protein) and a viral core containing a positive sense RNA strand. The S protein is responsible for host specificity and host binding, an essential step for the injection of the viral core into the host cell. The N-terminal S1 domain of SARS-CoV-2 has a high affinity to the membrane bound human Angiotensin-II Converting Enzyme (ACE2) protein allowing the virus to adhere to the cell surface exposing the S protein to host proteases to initiate infection. This mechanism is shared by several known human pathogenic coronavirus [6,7,8,9]. Furthermore, the S1 sub-unit has high genetic variability among coronaviruses, allowing these viruses to cross-species and thereby highlighting the threat of coronavirus in the future [7,10]. The binding affinity of viral S protein to ACE2 implicates ACE2 as a drug target against SARS-CoV-2. Thus, interrupting the interactions between S1 and ACE2, either through competitive or allosteric inhibition, is of interest as a preventive treatment.

ACE2 is a metallopeptidase that cleaves hormonal peptide angiotensin II at the carboxylic terminal phenylalanine and hydrolyses it to a vasodilator, angiotensin (1–7) [11]. Furthermore, it also shows peptidase activity against bradykinin, apelin, neurotensin, dynorphin A, and ghrelin, playing a crucial role in the regulation of several hormonal pathways [12]. Specifically, ACE2 belongs to Renin-Angiotensin-Aldosterone System (RAAS) where Angiotensin I Converting Enzyme (ACE or ACE1) converts angiotensin I to angiotensin II, a potent vasoconstrictor which in turn is converted into angiotensin (1–7) by ACE2. This system is tightly regulated through orchestration from liver, lungs, kidneys, and renal gland [13]. Thus, it is of the outmost importance to be able to avoid the viral infection while preserving the biological function of this enzyme. To achieve this task, the ACE2:S1 binding interface or the S1 fragment would represent suitable targets. However, protein-protein interaction surfaces are largely featureless, with no direct clefts or pockets amendable to small-molecule recognition. Furthermore, examination of the S1 surface involved in ACE2 binding demonstrates this problem for Spike. The S1 surface is featureless, relatively structurally smooth with no obvious pockets or clefts for small molecules to bind to with high affinity. Small molecules that modulate ACE2 conformational dynamics related to its enzymatic function can be useful tools to modulate CoVID-19 pathology as well as potentially regulating the RAAS system, expanding this study beyond the scope of CoVID-19 treatment.

Recently, Yan et al. [14] has published the structure of ACE2 in three conformations: an open conformation, a closed conformation, and a closed conformation in complex with a fragment of the viral S protein (Figure 1). The ACE2 open and closed conformations differ from each other by the degree of opening of the catalytic site cleft of the peptidase domain (Figure 1b) [14,15]. This causes a distortion of the ACE2:S1 binding interface [15]. Moreover, ligand binding studies have identified a closing motion associated with ligand binding, suggesting that the closed conformation represents the catalytically active state of the enzyme [16]. These structural insights imply ACE2 as a viable target to block S1 recognition through allosteric control of open-closed transitions necessary for S1 recognition. Indeed, several groups have performed computational drug screens on libraries of approved molecules for potential therapeutic targets and rapid deployment by targeting ACE2 as well as proteases that initiates membrane fusion [17,18]. We note that the previous computational drug screens of the ACE2-Spike complex focused on limited size data-sets, considered only a single ACE2 structure, and were limited to the ACE2-Spike interface [19,20]. Such an approach is not able to identify potential allosteric inhibitors of ACE2-Spike complex formation, leverage potential structure-based mechanisms of drug action, and may miss higher affinity sites within the ACE2 protein. Nevertheless, under the guidance of computational studies taken from both pre-print as well as peer-reviewed papers, in vitro assays and in some cases clinical trials have been performed on various pre-approved drugs [21,22,23,24,25,26,27,28]. As such, computational studies have been of tremendous assistance, however, most of these screens can only be performed on limited-size libraries of FDA approved compounds for swift drug deployment, which precludes gathering information on mechanistic models as well as de novo drugs with potential high binding affinities for a long term development.

With this study, we provide a computational strategy that leverages the ability of our machine learning algorithm, SSnet, to rapidly screen a vast amount of compounds. On top of aiding the protocol for immediate drug repurposing, using our model we developed a platform that can be used intuitively and quickly to aid de novo drug design. Specifically, with this study, we demonstrate the efficacy of such protocol, both in terms of accuracy and speed, by identifying existing FDA approved drugs that may treat CoVID-19 and possible mechanisms of action, including allosteric inhibition. These strategies can be leveraged to expand to databases on the scale of millions of compounds, lending the approach amendable to the development of de novo treatments. To demonstrate the efficacy of the approach, we first screen a library of approved drugs from DrugBank and ZINC using a fast and accurate machine learning approach termed SSnet to identify compounds predicted to have high-binding affinities [29]. These hits are cross-validated against traditional drug docking algorithm smina using the Autodock Vina scoring function [30]. The SSnet approach is then extended to a library of 750,000 in BindingDB, to discard compounds that are predicted to have a poor capacity for binding. The truncated library can then be assayed using alternative drug docking approaches and subsequent analysis using molecular dynamics, as well as in vitro assays, to identify possible targeted therapeutics (Figure 2). The analysis and interpretation of the results of the screening large datasets represent a challenge. For this purpose, we built a web interface for easy and intuitive access of our results to provide a platform for the identification of molecular scaffolds and functional groups that might influence target binding, thereby making the results accessible to a broader audience.

## 2. Materials and Methods

### 2.1. Dataset

We choose three datasets for screening: (i) Approved: Clinically approved list of drugs or natural products (FDA or world) compiled from DrugBank and ZINC databases, (ii) Natural: South African based herbal medicines SANC [31] and Brazil based herbal medicines NuBBE, [32] (iii) BDB: the BindingDB [33] dataset that has a large number of compounds already been tested to have activity with protein target.

### 2.2. SSnet

SSnet is an end-to-end based deep neural network framework that takes a protein structure and ligand information to predict protein ligand interaction (PLI) probability. The protein structure is taken from PDB formatted file which is used to extract curvatures κ and torsion τ of the protein backbone. The ligand is taken as Simplified Molecular-Input Line-Entry System (SMILES) string, which is utilized to extract the Morgan Fingerprint [34] of the ligand. The curvatures and torsion of the protein provide unique patterns due to multiple atomic interactions including side chain interactions, which play a major role in PLI. This information is used by SSnet to score the likelihood of ligand binding to target protein at IC50 less than 10 nM. SSnet model outperforms state-of-the-art machine learning models like Atomnet, [35] 3D-CNN, [36] and GNN-CNN [37] and classical force field and knowledge based methods employed by Autodock Vina [38] and Smina [30] in identifying positive protein-ligand pairs (protein ligand complex with high binding affinity). Since SSnet is a pre-trained algorithm [29], it can rapidly screen drugs from a large database of drugs to find positive hits on the target protein. SSnet is made available for public use on https://github.com/ekraka/SSnet with instructions for both training the neural network as well as calculating scores using the BDB trained model. The resources utilized and the speed of execution for SSnet are provided in the Appendix A.

### 2.3. Ligand Preparation for Docking

Ligands were obtained from respective datasets in SMILES format which was then converted to 3D structures using openbabel’s generate 3D option. The quality of the structures was checked via a python script using atomic distances as a criterion. 3D structures for the faulty structures were regenerated using rdkit [39]. A known limitation of docking methods is dealing with ligands containing a large number of atoms (high degree of freedom) [40]. Approximately 300 structures were excluded from the list of approved and natural compounds since neither rdkit nor openbabel were able to generate 3D structures of the compounds with a high degree of freedom (n > 50) which are provided in the supporting information). Explicit polar hydrogen atoms were added using openbabel. Lastly, the generated 3D structures were converted to pdbqt format.

### 2.4. Virtual Screening and Docking

To perform virtual screening and docking, smina was used on a subset of clinically approved drug lists obtained from dataset 1. ACE2 structures from PDB ID 6M1D, 6M17, and 6M18 were taken as well as viral S protein fragment in 6M17. As such, chain B corresponding to ACE2 and chain E for viral S protein fragments were extracted from the structures using pymol with Zn in the catalytic site [41]. The smina runs were performed on the prepared ligands by centering the box of size 32Å × 32Å × 32Å around Lys353 (Appendix A), identified as a key interaction residue between ACE2 and S protein, [19] with exhaustiveness of 36 on the default scoring function. The screens were replicated three times to consider the variability of scores from smina. We also re-evaluated the top 100 scoring ligands using exhaustiveness of 504 in triplicate. The results did not significantly impact the affinity scores and had only a weak effect on the standard deviation across the three replicates. As such, we did not increase the exhaustiveness for the entire compound library.

### 2.5. Chemical Sorting

H${}_{2}$O was considered as the first entry for the list of compounds. The list was then sorted recursively such that each molecule is most similar to the previous molecule in the indexed list. The molecular similarity was considered using Tanimoto Coefficient (TC) of extended circular fingerprint (ECF, specifically Morgan Fingerprint). Approved and natural datasets were sorted in this manner, however the size of BDB dataset limits the generation of such a list. Thus, the BDB dataset was sorted by using k-nearest neighbor algorithm to consider 10,000 nearest neighbors computed by TC. The most similar compound in the 10,000 nearest neighbors was selected as the next molecule in the indexed list. Starting from H${}_{2}$O, this process was performed until every molecule was sorted. In case of no neighbor within 10,000 presorted compounds is found, the most similar compound is computed back from all compounds of the BDB dataset excluding the sorted entries. The benefit of using such an algorithm is to compute all nearest neighbor for each compound via parallel computation, significantly reducing the computation time, however, does not guarantee that the two consecutive compounds are most similar to each other from the remaining list. For use in the grouping, to display the data, this approach works despite the pitfall.

### 2.6. Data Visualization

The sorted list was used as an index for a line which was then mapped to a pseudo-Hilbert space filling curve. The order of the pseudo-Hilbert curve was taken such that the compounds would be represented by at least one line segment in the curve. A detailed description of Hilbert space filling curve is discussed in the supporting information. The pseudo-Hilbert curve was used to represent the compound list for three reasons: (a) high density of data can be displayed in high resolution, (b) the pseudo-Hilbert curve preserves the spatial proximity of the line onto the map, and (c) similarity sorting of compounds allows easy identification of clusters of molecular scaffolds directly from the map. The preservation of spatial proximity ensures that the sorted list of SMILES based on the Tanimoto Coefficient (TC) is represented in the 2D space by spatial proximity. This representation allows immediate identification of clusters of molecules. The curve was then colored based on the SSnet binding prediction or smina affinity scores. A website was created to allow interactive exploration of the map to identify scaffolds of compounds that are scored highly either by smina or SSnet. The website is made available for public use at https://CoVID19screen.smu.edu/.

## 3. Results

Computational approaches to screening large molecular databases can be limited due to the computational time required to exhaustively search the conformational space of small molecules that determines diverse binding modes to protein targets. To improve computational efficiency, we have employed a tiered approach to identify potential ligands that bind to the ACE2 receptor and possibly function as CoVID-19 treatments. We target two conformations of ACE2 receptor, open and closed, as well as ACE2 in complex with S1 (closed) Figure 1. S1 from the S protein of SARS-CoV-2 did not co-crystallized with ACE2 (open) as observed by Yan et al. [14], thus we posit that S1 cannot bind ACE2 in its open conformation [14].

To that effect, we first test the validity of SSnet prediction scores in two ways: (1) We compare the prediction score for ACE2 against ACE1, two members of the same protein family. (2) We compare SSnet scores to binding affinities computed by smina for a small library of FDA and World approved drugs for which docking method is feasible. For the first, we observe a difference in scores for ligands between ACE2 and ACE1 as seen by the deviation for y = x line on Appendix A. These results indicate that SSnet can differentiate from closely related proteins and is not biased to specific tertiary or domain folds. For the second, we performed a closer analysis on the results described in SSnet Predicts Ligands with Low Smina Binding Affinities. Upon validation, we can proceed with using SSnet to rapidly screen through a large database of drug-like compounds.

### 3.1. SSnet Predicts Ligands with Low Smina Binding Affinities

SSnet, being a machine learning model, is not free from the pitfalls of overfitting. Thus, a crucial task is to first check how SSnet correlates with other methods such as smina. The results of screening for SSnet and smina scores over all approved drugs (8000 compounds) are shown in Figure 3. The result from screening these compounds demonstrate a strong correlation between smina and SSnet (Figure 3). Briefly, high binding probability hits from SSnet have corresponding low binding affinities computed by smina. We observe that for 1097 ligands with a prediction score of 0.5 or higher, only 7 ligands (0.6%) have higher than −6 kcal/mol binding affinity in smina. These values demonstrate that SSnet has a more stringent acceptance rate than smina and thus a higher likelihood of finding true active ligand from a large pool of molecules.

### 3.2. Ligands with High Binding Affinity Scores

Molecules that demonstrate high scores in both SSnet and smina are prime candidates for investigation. Preliminary examination of heat maps predicted by SSnet, and poses identified by smina, indicate that compounds bind to two sites depicted in Figure 1. The majority of compounds bind to the ACE2 catalytic site proximal to the S1 binding interface. As discussed below, we do observe some molecules that directly bind to the interface, or transect the ACE2 catalytic site to contact the S1 interface. In both cases, these ligands could impact S1 recognition through either allostery or direct competition. Although these screens might not reflect the efficacy of drugs in vivo, these results allow us to focus on a handful of ligands that can be experimentally tested. In Appendix A, we complied a list of 12 drugs recognized by both smina and SSnet to be strong binders as well as drugs that are predicted by one and not the other. The top scorers are taken for all the proteins together to highlight some of these top predicted binders.

#### Top Scores for both Smina and SSnet

Our combined machine learning and docking strategy return high binding affinity ligands consistent with previous computational screens of the ACE2 receptor. Furthermore, sorafinib, [42] irinotecan, [43] and nilotinib [44] (Appendix A) show reduction in infection rates in cell assays studies, while zanubrutinib is currently in clinical trial. Although these drugs target other pathways involved in the pathology, the reduction in infection rate could be in part attributed to the ability of the drug to bind ACE2. MD, simulations and drug assays specifically targeting ACE2 and ACE2:S1 would be required to test the hypothesis.

Examination of the binding locations for compounds that are either currently being tested as CoVID-19 treatments, or may intersect with ACE2/RAAS indicates that the compounds primarily bind to the ACE2 catalytic site (Figure 4). Although the highest affinity poses of indinavir, zanubrutinib, and sorafenib all reside in the catalytic site, sorafenib has a pose of comparable score within the ACE2-Spike interface (Figure 4c). In either case, the compounds contact key elements involved in S1 recognition, and could impact ACE2-S1 interactions.

### 3.3. Top Scores with SSnet

An examination of compounds identified by SSnet to have a high probability in binding identifies they primarily fall into three pharmaceutical categories, antivirals, protease inhibitors, and kinase inhibitors (Appendix A). Venetoclax [45] and Aliskiren [46] have shown efficacy towards CoVID-19. Since ACE2 hydrolyzes the peptide hormone angiotensin II at C-terminal phenylalanine as well as multiple additional regulatory peptides [11], protease inhibitors could potentially bind to and inhibit the catalytic site. Consistent with such a mechanism, examination of the binding poses of antiviral compounds reveals primary binding within the ACE2 catalytic site, although some antivirals do dock directly to the ACE2-S1 interface in some poses (Appendix A).

In addition to antivirals, and protease inhibitors, SSnet recognize among the top scorer an opioid and linaclotide, used to treat irritable bowel syndrome. Most of the anti-cancer drugs have substantial side effects, and most of them, with the exception of afatinib, have reported side effects correlated with alteration of blood pressure. This might be an indication of the correlation of this anti-cancer drugs and ACE2 related biological pathways. It is worth noting that all the drugs identified as top scorers by SSnet are all good binders according to smina binding affinity.

### 3.4. Model of Drug Action

Protein-protein interactions require the proteins involved to be in specific conformations. The optimal conformation for protein-protein binding can be both induced or inhibited by drug binding. While competitive binding of ligands at the protein-protein interaction interface can directly inhibit the interaction, ligand binding in allosteric sites can stabilize conformations that inhibit or enhance the protein-protein interaction Yan et al. [14]. observed that the viral S1 fragment could only be co-crystallized with ACE2 in the closed ACE2 conformation. In contrast, in the absence of S1 fragment, ACE2 crystallized in both open and closed conformation [14]. Their results indicate that the allosteric control of ACE2 conformational dynamics may enable a robust way to block ACE2:S1 recognition and subsequent SARS-COV-2 infection. Since SSnet predicts the same binding affinities and binding regions regardless of the protein conformation, when multiple structures of a protein are available, SSnet is able to predict ligand binding using a single conformation. Further, PLI prediction can be performed for both ACE2 and ACE2:S1 to investigate selectivity. This approach allows us to propose a hypothesis for PLI mechanism. Herein, we propose four scenarios for ACE2-S1-ligand interaction, summarized in bullets below and shown in Figure 5.

Case-I: The ligand binds to the open conformation but does not prevent the protein from exploring the closed conformation. This would render the drug ineffective.Case-II: The ligand binds to and stabilizes the closed conformation. This may make the drug counter-productive by making the receptor more susceptible to the docking of the viral S protein.Case-III: The ligand binds to and stabilizes the open conformation. This would prevent the docking of the viral S protein since the closed conformation is no longer explored.Case-IV: The ligand binds to the closed conformation with or without the viral S1 protein and biases the receptor towards an open conformation. This would either prevent or disrupt the viral docking.

Previous validation of SSnet indicated the method was robust in identifying both allosteric regulators, and ligands that bind to hidden conformations. In this regard, it is well-suited to identify potential regulators for case I–IV. Further, our rapid method can be implemented to screen large drug libraries against all three structures and together with secondary screening in smina, can be used to select for molecules that potentially fall into the favorable Case-III and Case-IV scenarios. The mechanisms of drug action proposed cannot be validated using drug docking and scoring methods alone and will require methods that apply higher levels of theory like molecular dynamics or experimental verification. However, we can still screen for Case-III as the drugs that present a high score for the open conformation but low for the complex might lead to the stabilization of the open conformation and thereby disrupt spike recognition. The case-III would apply for both interface binding ligands as well as ligands that bind to the catalytic pocket or peripheral sites not examined in previous CoVID-19 computational screens. Using these criteria, Appendix A was generated to seek drugs that may selectively bind to ACE2 to negate interactions with S1 via allosteric disruption of ACE2. In addition to the scenarios just presented, the virus could be prevented from binding ACE2 by drug molecules that bind to the surface of the peptidase domain of ACE2, which is the interface where the viral S1 spike protein subunit binds.

### 3.5. Top Scores for SSnet ACE2 (Open)-ACE2:S1 (Closed)

Following the rationale illustrated in our proposed allosteric mechanisms of drug action, where a drug that binds preferably to the ACE2 open conformation compared to the closed one might stabilize the open conformation, in Appendix A the scores from the SSnet scores based on the difference between ACE2 in the open conformation and ACE2:S1 complex in closed conformation are presented. Pyronaridine [47], have shown efficacy towards CoVID-19 in cell assays while Methylprednisone, Linagliptin, and ormeloxifene are in clinical trials (Appendix A). The top smina docking poses for compounds with a high selective difference (SD) indicates that they lie within the ACE2 catalytic site, close to ACE2:S1 interface. Antivirals such as Nelfinavir [22], and remdesvir [21] are in clinical trials which all display high SD. Remdesvir demonstrates a preference for the open conformation (SSnet = 0.748) compared to the ACE2:S1 complex (SSnet = 0.578) with a selectivity difference(SD) of 0.17. Lopinavir and ritonavir have shown SD of 0.08 and 0.09 respectively with a preference to open conformation of ACE2 with SSnet scores of 0.90 and 0.91 respectively. Although not presented in the table, we highlight these molecules since they have high SSnet scores as well as are in the clinical trial. These results provide a favorable outlook on the methodology proposed here and might indicate a complementary route of actions of these drugs that leverage the conformational selectivity of the viral spike protein.

### 3.6. Zinc Effect on SSnet Binding Probabilities

ACE2 is a zinc-dependent metalloprotease, which cleaves the C-terminal residue of angiotensin II within its catalytic site. Given the importance of zinc for function we deemed it necessary to include zinc for all smina docking runs. Notably, SSnet was originally trained on structural models independent of any bound metals or cofactors. As such, SSnet strips protein targets of all ligands and cofactors, including zinc. Although SSnet has proven robust in identifying high affinity drug molecules, including for metalloprotein targets, we decided it was important to examine if the presence of zinc may alter predicted affinities from SSnet. To force SSnet to consider ligands in the presence of zinc, smiles strings were modified to include zinc in a manner that mimics co-analysis (or co-administration) of zinc and each ligand. Noteworthy is the fact that SSnet considers metal binding to the protein. Further, SSnet can recognize multiple ligands such as Ferrous cysteine glycinate (Fe${}^{2+}$ and cysteine glycinate) and it is therefore capable to consider both Fe${}^{2+}$ and cysteine glycinate together. Comparison of the SSnet scores with and without zinc revealed unexpected results that may impact analysis of compounds selective for the open and S1-complex structures Figure 6 and Appendix A. Correlations of SSnet with and without zinc, indicate the inclusion of zinc has only a modest positive effect for most compounds. This is especially true at low and high scores, where the two methods are equivalent. In this manner, the two methods do not differ in the selection of top-binding compounds. Based on a preliminary analysis, we observed that the top 15 ligands influenced most strongly by the inclusion of zinc lie in the catalytic site near the zinc ion binding site (Appendix A). Further examination reveals two aspects, which may have implications in the identification of compounds selective for the open conformation. First, the effect of zinc on SSnet scores is significantly more pronounced in structures lacking S1 (Figure 6b). In this regard, zinc seems to enhance ligand binding in the absence of the S1 complex regardless of whether ACE2 is in the open or closed conformation. Second, the effect of zinc is most pronounced within the range of modest SSnet scores where we observed the most selective compounds. To determine whether the role of zinc in SSnet scores may be artifactual, we sorted the approved compound libraries by the difference between SSnet with zinc and SSnet without zinc.

Examination of compounds with the largest SSnet score differences with and without zinc are summarized in Appendix A. Here, we observe members of the proton pump inhibitor (PPI) families. Notably, PPIs are well known to potentially cause zinc deficiencies, due to the high affinity of PPIs to zinc. Analysis of other members observed in the top 15 affected compounds, indicates that all are known to have high affinities for zinc, including an ACE-inhibitor and ACE2 receptor blocker. These results strongly suggest that zinc may preferentially enhance binding to the ACE2 receptor that may be particularly relevant to compounds selective to the uncomplexed vs. ACE2-S1 complex structures. Based on these results, we tabulated the top compounds with SSnet(Zn) scores based on the difference between ACE2 in the open conformation and ACE2:S1 complex in the closed conformation in order to identify any that may be zinc dependent (Appendix A). Notably, three estrogen-related molecules, estriol-3-glucuronide, estradiol glucuronide, and 17-alpha-estradiol-3-glucuronide are now observed in the top 20 most selective compounds.

### 3.7. Natural Compounds and Large Drug Molecules

An interesting finding of our smina screening is the abundance of natural compounds from the NuBBE and SANC datasets in the top binder list (Appendix A). We see compounds related to avonoids, flavonones, and polyketides in this group. We note that several glycopeptide antibiotics and macrolactams also demonstrate high affinities in smina. However, the comparison of smina affinity scores for these large molecules is complicated by poor 3D structures in molecular databases. Further, high degree of flexibility and conformational degrees of freedom limits the ability to identify the highest affinity pose, as demonstrated by a high average of absolute deviations for some of these compounds. Due to these constraints, we identified that smina has a limitation in the reproducibility of docked energies for these large molecules. They are not included in the tables above but are discussed in the discussion below. Despite the limitations of smina in analyzing these molecules, we do not exclude them entirely for two reasons. First, compounds in these families such as azithromycin and oritavancin have been suggested as CoVID-19 treatments. Further, the efficacy of these large molecules for the intended targets is very high with the working concentrations at sub nM range [48,49,50].

### 3.8. SSnet as Tool to Aid the Discovery of De Novo Compounds

In order to assist the design of de novo drugs tailored to fight SARS-COV-2, we developed an interface where libraries of non-approved compounds can be clustered based on structural similarity and visualized in a 2D heatmap based on their affinity and selectivity to a particular conformation of the ACE2 receptor. Potentially active molecular scaffolds can be inferred from the visualization of high affinity compound clusters and used as starting point for further drug design. SSnet proved to be sensitive to functional group modifications. This translates into direct valuable information for drug designers which are now provided not only with potential active scaffold but functional group information to build pharmacophores and help rational drug design as well. An example is provided by estradiol. When analyzing Estradiol analogues from the heatmap derived cluster, they showed considerable deviations in binding affinity upon functional group substitution as can be seen in Figure 7.

Hormones are extremely selective in their functionality and as such small changes in functional groups could significantly alter their selectivity towards protein target. Since SSnet has been trained on a wide variety of ligands from BDB, it is not surprising that the selectivity of hormones is well captured by our model. This is demonstrated in Figure 7, the coordinates for estradiol were selected in the pseudo-Hilbert map on the website to identify other potential estradiol derivatives that could have strong binding to ACE2. It is important to mention that a normal screening values-based sorting would have not provided such insight as estradiol derivatives are not in the high scoring area. Ulterior information from the clustering on the heatmap can be used to extrapolate how small groups affect the selectivity towards the ACE2 or the ACE2:S1 complex. Figure 8 shows that the highest selectivity is achieved for ACE2 binders with binding probabilities in the range 0.60–0.70. Furthermore, SSnet is blind to tertiary protein conformations, with only up to 0.5% difference in drug scores between the ACE2 open and closed conformations (Figure 8c). This characteristic reduces the workload since only one ACE2 conformation and one ACE2:S1 complex conformation are needed. Being able to avoid the screening against all the crystal structures allowed us to save precious computational resources and time in the screening of 750,000 compounds in the BindingDB database.

## 4. Discussion

The pressure to rapidly find a cure of preventive treatments in response to the life and economic costs of the CoVID-19 pandemic has exposed the necessity for fast and reliable protocols and/or methods to identify treatments when new pathogens appear. The existing computational studies being performed to find therapeutic for CoVID-19 showed that the current approaches have three main limitations: (1) Most studies have used a low-level of exhaustiveness in molecular docking algorithms. Such an approach can lead to difficulty in finding local minima in receptor ligand interactions are a high degree in variability in resulting binding affinities. (2) Computational efforts have been limited to a distinct conformation of the ACE2 receptor and compounds that bind to the ACE2-Spike interface. Such an approach does not leverage the three known structural states of the ACE2 receptor, which include an open conformation that cannot crystallize in complex with the Spike protein [14]. As such, compounds that preferentially bind to the open or closed conformations that may stabilize Spike-incompetent binding states are missed. (3) The approaches are computationally intensive, which limits the utility of the approach in large compound libraries for de novo drug identification. To alleviate these issues we have developed a robust platform to rapidly screen large compound libraries, in three different states of the ACE2 receptor, to identify both existing approved drugs and de novo drug candidates.

With this study, we showed that our protocol that leverages the recently developed PLI prediction ML algorithm, SSnet, can greatly reduce the chemical space of molecular libraries and, at the same time, ensuring that potential tight binders are not excluded. The fact that SSnet is blind to protein conformation allows us to consider all cryptic and allosteric sites by only looking at a single structure. Our PLI prediction using SSnet against the ACE2 structures to find potential treatment for CoVID-19 resulted in identifying potential binders that are being confirmed experimentally as a treatment for infected individuals.

It is interesting to evaluate how our protocol composed of a tertiary screening, SSnet, and secondary screening, smina, ranked FDA approved drugs. Multiple molecules identified in this study emerged in other computational studies, have been investigated using in vitro and in vivo assays, and even gone through clinical trials. Antivirals such as remdesvir, nelfinavir, and lopinavir-ritonavir are prominent examples of drugs under clinical investigation. Interestingly, these molecules selectively bind to the virus-incompetent ACE2 open conformation, giving an indication of the validity of our model of drug action. We found that hormones, and in particular estriol metabolites are amongst our tightest binding compounds in smina, and have SD scores exceeding 0.20 for SSnet. In the case of all hormones, we observe that they are binding adjacent to the Zn${}^{2+}$ binding site required for the function of ACE2, and the primary site differentiating the open and closed conformations. Notably, previous computational screens neglect this site, as it is not directly in the ACE2:spike interface, however, the site does modulate conversion between the open and closed conformation that is believed to be necessary for Spike recognition. These results may add utility to our workflow as a robust means to quickly identify ligand scaffolds as well as pre-existing drugs with high binding probabilities. Currently, the sex-dependent difference in CoVID-19 pathology and therapeutic effects of estrogen are being associated with different ACE2/TMPRSS2 expression, [51,52,53,54,55,56,57,58,59], or immune system modulation [51,55,56,60,61,62,63]. To the best of our knowledge direct binding of sex hormones to ACE2 has not been observed. Given their presumptive ability to selectively bind to the open conformation, sex hormones could have an additional role in disease progression and pathology. Lastly, we did observe an abundance of glycopeptide and macrolactam antibiotics within the top scoring compounds in SSnet and/or smina. We caution in over-interpreting smina results from this class of compounds as the high-degree of freedom leads to difficulties in obtaining consistent high-affinity poses. Issues with large flexible molecules is a known limitation of existing computational docking algorithms [64,65]. The ability of SSnet to circumvent these issues and selectively identify high probability binders demonstrates the efficacy of using SSnet as a front-end to drug discovery.

Furthermore, SSnet was shown to be able to emulate co-administration. Based on the requirement of bound zinc for ACE2 function, we forced SSnet to examine ligand binding in the presence of zinc ions. Analysis of molecules affected by this approach identified known zinc-chelators as having significantly affected SSnet scores, that impacted selectivity for the open conformation. We note, that the approach used to force SSnet to examine the role of zinc ions mimics co-administration of Zn and some FDA approved drugs. The results indicate that co-administration of zinc may enhance interactions between some FDA approved drugs and the open conformation of the ACE2 receptor.

Extending the limits of previous computational studies we performed a screening using only SSnet on a large database of BDB (750,000 ligands) to obtain useful information regarding de novo drug design. As described in the result section, we developed a web interface where molecules are clustered based on similarity on a 2D map and colored based on binding affinity to the protein. By selecting points into the interface the user can further explore the effect of different scaffolds and functional groups on the binding score or affinity. With our web-based tool, we aim to provide a fast, intuitive, and flexible way to aid the discovery and design of CoVID-19 therapeutics as well as other diseases. To the best of our knowledge, such an approach for visualizing the chemical space for drug discovery has not been done. This approach can be used for other therapeutical targets besides CoVID-19.

With this study, we highlighted how SSnet can represent a powerful tool in a computational drug discovery protocol. However, in order to obtain crucial information needed for understanding the model of drug action and lead optimizations, other tools have to be used in cooperation with SSnet. In fact, while the latter can provide a fast and reliable first-screening of large libraries, it cannot provide binding poses, needed to analyze protein-ligand interactions, and binding affinities, needed for scoring the most promising drug candidates. The screened results can be directly employed for rapid testing using assays when urgency is required. However, from a drug design perspective, drug docking, molecular dynamics simulations, and binding free energy calculations are still necessary. Lastly, if a drug discovery campaign is aimed at a specific conformation of a protein, SSnet will include in the top binders drugs that bind to alternative conformations of the protein as well.

## 5. Conclusions

Herein, we have developed a rapid screening method suitable to compound libraries on the scale of millions of compounds that is capable with high accuracy to identify probable high affinity compounds for two conformations of the ACE2 receptor and the ACE2-S1 complex. These compounds can function as putative leads for de novo drug discovery. We have further developed a web interface to allow researchers to rapidly identify high affinity scaffolds for in vitro characterization and structure-function activity studies. We believe that an open-access utility such as this will allow diverse researchers to contribute to the discovery of both existing FDA approved drugs and de novo development of CoVID-19 treatments. Further, the methodology can be easily extended to any protein, or protein complex involved identified as putative drug targets.

## Figures and Tables

**Figure 1 ijms-22-01573-f001:**
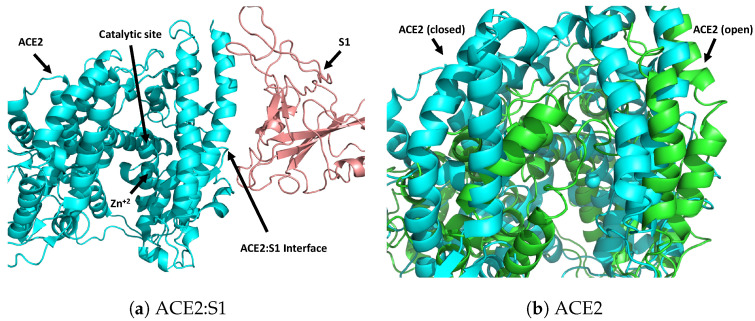
Crystal structures showing interaction between human ACE2 and S1 domain of viral S-protein. (**a**) Viral S1 fragment of the S-protein co-crystallized with ACE2. Only closed conformation can be co-crystallized suggesting ACE2 conformational dependency for S1 interaction [14]. (**b**) Closed and Open conformation of ACE2. Zn cation is not detected in the open conformation. ACE2 freely explores both open and closed conformations [14].

**Figure 2 ijms-22-01573-f002:**
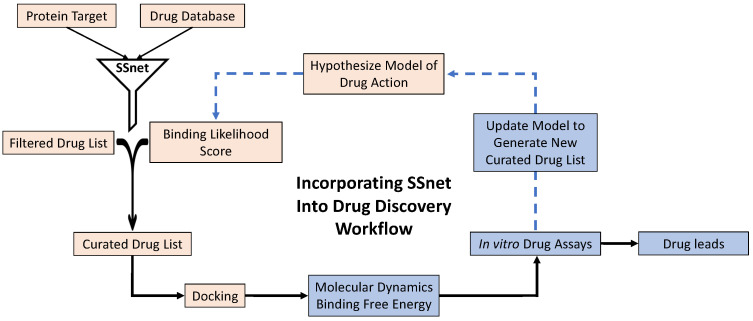
Proposed drug discovery workflow using SSnet. The orange boxes represent the steps that are performed in the present study. The blue boxes represent further steps required to complete the drug discovery workflow to obtain drug leads.

**Figure 3 ijms-22-01573-f003:**
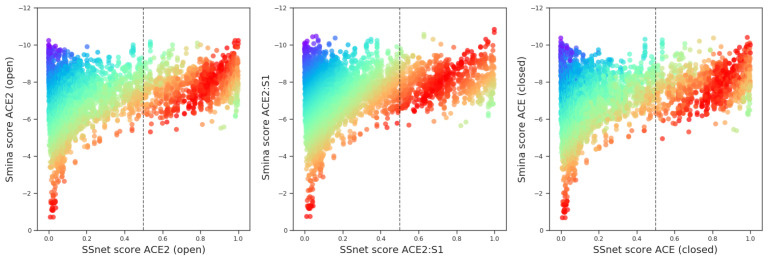
Correlation map between smina binding affinities and SSnet scores. Red color indicates strong agreement between the two methods. Blue color indicates strong disagreement between the two methods.

**Figure 4 ijms-22-01573-f004:**
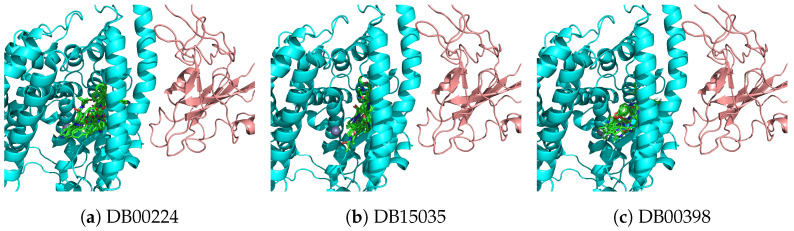
Top 9 lowest energy poses for compounds with high scores in both SSnet and smina of biological interest to CoVID-19. Panel (**a**): Indinavir binding poses. Panel (**b**): Zanubrutinib binding poses. Panel (**c**): Sorafenib binding poses.

**Figure 5 ijms-22-01573-f005:**
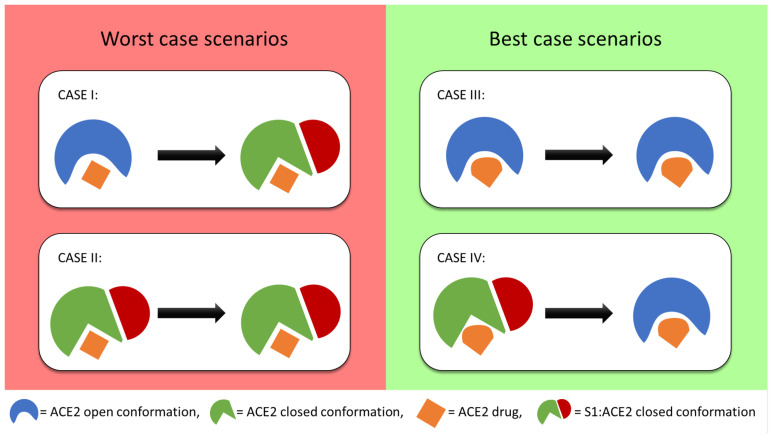
Proposed mechanism of drug action for allosteric inhibitors of ACE2:S1 binding. Case I and Case II represent undesired stabilization of the ACE2:S1 complex. Case III and Case IV represent ACE2:S1 complex inhibitors due to allosteric disruption of the ACE2 binding interface, resulting in the stabilization of the open ACE2 conformation.

**Figure 6 ijms-22-01573-f006:**
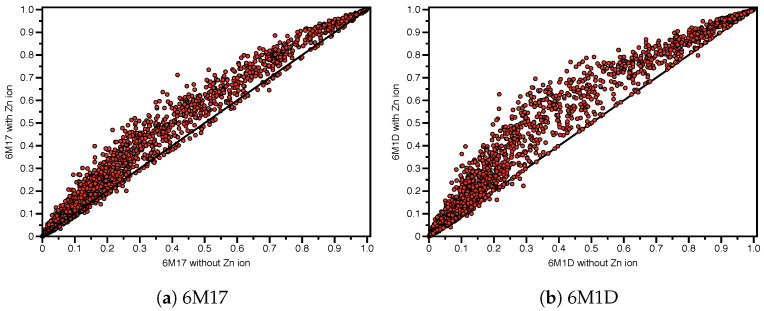
Effect of zinc ion on SSnet binding probabilities.

**Figure 7 ijms-22-01573-f007:**
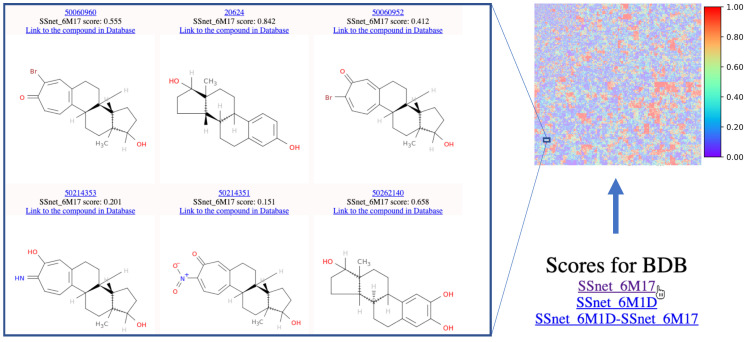
A road-map for navigation of the chemical map to find areas of interest. The cluster of molecules with high similarity to Estradiol, when the latter was selected in the map, are shown.

**Figure 8 ijms-22-01573-f008:**
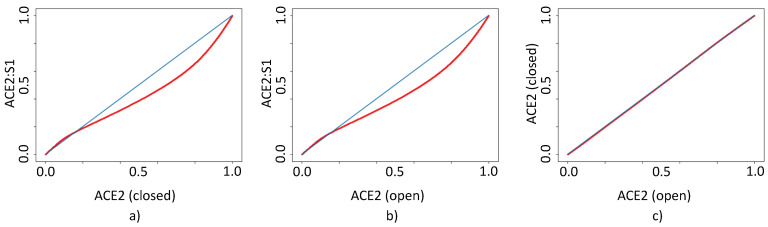
Correlation diagram for SSnet scores between each of the protein targets for ligands in BDB (**a**) ACE2:S1 vs. ACE2(closed), (**b**) ACE2:S1 vs. ACE2(open), and (**c**) ACE2 (open) vs. ACE2 (closed). The red curve shows the scores for ligands plotted as x, y. The blue curve is an x = y line representing non-selectivity to the ACE2 conformation. The highest difference between scores is achieved for probabilities in the range 0.60–0.70. High probability binders show no selectivity.

## Data Availability

SSnet is openly accessible for both training and prediction at SSnet-GitHub (https://github.com/ekraka/SSnet). The website developed in this work is live at COVID-19Screen (https://CoVID19screen.smu.edu/).

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
