# Peer review of "Predicting Potential SARS-COV-2 Drugs—In Depth Drug Database Screening Using Deep Neural Network Framework SSnet, Classical Virtual Screening and Docking"

_ijms, 2021, doi:10.3390/ijms22041573_

Round 1

Reviewer 1 Report

This manuscript titled “Predicting Potential SARS-COV-2 Drugs - In Depth Drug Database Screening Using Deep Neural Network Framework SSnet, Classical Virtual Screening and Docking” mainly described the screening method developed by authors, in combination with docking. Authors screened approved drugs and compounds from the binding database. They identified the high-affinity compounds for more than one conformation of ACE2 protein.

Minor comments:

  1. Can authors describe the difference in open and closed conformations of ACE2 protein?
  2. Did the authors use only ACE2 specific compounds from the binding database?
  3. Did authors try a different way to convert molecules to avoid excluding ‘defective 3D structures’ (The list of molecules provided in supplementary)

Major:

  1. I suggest performing 100 ns MD simulations for at least the top 5 compounds they identified for validation of their model.
  2. Why authors are targeting the ACE2 receptor instead of Spike protein? How they can justify that ACE2 is a suitable target because ACE2 is involved in important functions.

Reviewer 2 Report

This article is well written. I only have minor concern regarding some grammar and spelling errors. I would suggest the authors to recheck the paper and correct any error.

Author Response

We have rechecked for grammar and spelling mistakes and updated the manuscript accordingly.

Reviewer 3 Report

This study proposes a method for screening the potential SARS-COV-2 drugs in different drug databases. It is a comprehensive method using deep neural networks, virtual screening, and docking. Although there are a lot of efforts to conduct the experiments, there still has room for improvements as follows:

1. There are a lot of previous studies to predict SARS-COV-2 drugs using machine learning and virtual screening with high performance. Even currently it has been found that some drugs have proven their efficiency in SARS-COV-2 treatment. Therefore, the contribution of this study is limited and not quite novel.

2. The authors should have some validation data on the models.

3. A big question is that the authors mentioned the use of SSNet, but I'm not sure whether the SSNet is a well-known method in this field or the method that the authors proposed by themselves. If the authors aimed to propose a new method, they need to mention it in more detail in the methodology (how they generated the methods?)

4. The authors used deep neural networks in the model, so which measurement metrics and evaluation methods that they have tried?

5. Deep neural network has been used in previous works related to bioinformatics such as PMID: 33260643 and PMID: 32613242. Therefore, it is suggested to refer to more works in this description.

6. How to select the optimal models?

7. What is the advantage of deep neural network in this dataset? As I was known, the main advantage of deep learning is to learn unstructured data or hidden features.

8. The authors should compare the predictive performance to previous works on the same dataset.

9. The authors should have spaces before the reference number.

Round 2

Reviewer 1 Report

The authors have addressed all the comments. This article can be published in the present form.

Reviewer 3 Report

My previous comments have been addressed.